# Integrative Machine Learning Framework for Enhanced Subgroup Classification in Medulloblastoma

**DOI:** 10.3390/healthcare13101114

**Published:** 2025-05-11

**Authors:** Kaung Htet Hein, Wai Lok Woo, Gholamreza Rafiee

**Affiliations:** 1School of Electronics, Electrical Engineering, and Computer Science, Queen’s University Belfast, Belfast BT9 5BN, UK; khein02@qub.ac.uk; 2Department of Computer and Information Sciences, Northumbria University, Newcastle NE1 8ST, UK; wailok.woo@northumbria.ac.uk

**Keywords:** medulloblastoma, 450 K methylation data, machine learning, subgroup classification, subgroup-directed therapies

## Abstract

Background: Medulloblastoma is the most common malignant brain tumor in children, classified into four primary molecular subgroups: WNT, SHH, Group 3, and Group 4, each exhibiting significant molecular heterogeneity and varied survival outcomes. Accurate classification of these subgroups is crucial for optimizing treatments and improving patient outcomes. DNA methylation profiling is a promising approach for subgroup classification; however, its application is still evolving, with ongoing efforts to improve accessibility and develop more accurate classification methods. Objectives: This study aims to develop a supervised machine learning-based framework using Illumina 450K methylation data to classify medulloblastoma into seven molecular subgroups: WNT, SHH-Infant, SHH-Child, Group3-LowRisk, Group3-HighRisk, Group4-LowRisk, and Group4-HighRisk, incorporating age and risk factors for enhanced subgroup differentiation. Methods: The proposed model leverages six metagenes, capturing the underlying patterns of the top 10,000 probes with the highest variances from Illumina 450K data, thus enhancing methylation data representation while reducing computational demands. Results: Among the models evaluated, the SVM achieved the highest performance, with a mean balanced accuracy 98% and a macro-averaged AUC of 0.99 in an independent validation. This suggests that the model effectively captures the relevant methylation patterns for medulloblastoma subgroup classification. Conclusions: The developed SVM-based model provides a robust framework for accurate classification of medulloblastoma subgroups using DNA methylation data. Integrating this model into clinical decision making could enhance subgroup-directed therapies and improve patient outcomes.

## 1. Introduction

Medulloblastoma is a type of primary Central Nervous System (CNS) tumor that predominantly develops in the cerebellum. It is the most common malignant brain tumor in children, representing 15−20% of all pediatric CNS tumors [1,2]. This tumor type is primarily diagnosed in children under the age of 15; however, it can also happen in adults and its incidence rate decreases with increasing age [3]. Each year in the United Kingdom, medulloblastoma is diagnosed at an incidence rate of approximately 0.37 cases per 100,000 person-years for individuals under 18 [4]. In the United States, the incidence rate among individuals aged 0–19 years is around 0.42 cases per 100,000 person-years [5]. The World Health Organization (WHO) classifies medulloblastomas universally as Grade 4 tumors due to their malignant and fast-growing nature. These tumors originate from fetal cells that persist after birth and can rapidly spread to other areas of the brain and spinal cord through the cerebrospinal fluid [6]. Given its aggressive nature and high prevalence in children, medulloblastoma remains a critical focus of pediatric neuro-oncology, requiring early detection and comprehensive treatment strategies to improve survival outcomes.

Traditional treatment strategies for medulloblastoma are tailored based on a combination of clinical and radiological risk criteria. Standard treatment typically involves surgery followed by radiotherapy, chemotherapy, and participation in clinical trials. The long-term survival rate for medulloblastoma varies significantly, averaging between 50– 90% depending on several factors, including cancer stage, tumor size, extent of surgical resection, patient age, treatment response, genetic profile, and overall risk level [2,7]. Since the 2016 update to the WHO Classification of Tumors of the CNS (fourth edition), molecular subgroup characteristics have been integrated alongside traditional histological variants [8,9]. This layered approach has refined treatment strategies by incorporating both molecular and histological features, offering more precise risk stratification and potentially improving patient outcomes. Survival rates and treatment strategies also differ significantly across these molecular subgroups, underscoring the importance of accurate molecular diagnosis.

A promising approach to classifying molecular subgroups of medulloblastoma involves DNA methylation profiling, which measures the methylation status of thousands of CpG sites across the genome. CpG refers to a specific DNA sequence where a cytosine (C) nucleotide is followed by a guanine (G) with a phosphodiester bond (p) linking the two nucleotides. Synthetic DNA sequences known as CpG probes are designed to bind to and detect the presence or absence of methylation at specific CpG sites. This technique identifies epigenetic changes associated with specific disease states or biological processes, collectively referred to as methylation patterns. These patterns, often detected using platforms such as the Illumina 450K or EPIC (850K) arrays, serve as precise biomarkers for distinguishing between different medulloblastoma subgroups.

In a study by Schwalbe et al. [10], researchers conducted DNA methylation microarray analysis, focusing on six metagenes—groups of genes whose methylation patterns are linked to biological processes. Through unsupervised class discovery, they identified seven novel molecular subgroups of medulloblastoma based on these metagenes. These subgroups represent further subdivisions of the primary molecular categories, refined by factors such as age and risk, each characterized by distinct clinical features and survival outcomes.

Building on this foundation, the objective of the present study is to develop an accurate supervised classification model, together with a software package tool, that uses these six metagenes to categorize medulloblastoma samples into the seven identified molecular subgroups: WNT, SHH-Infant, SHH-Child, Group3-LowRisk, Group3-HighRisk, Group4-LowRisk, and Group4-HighRisk. By leveraging these established methylation patterns, this approach aims to improve classification accuracy, enable more precise differentiation within subgroups, and ultimately support the development of tailored therapeutic strategies to enhance patient outcomes.

This paper is structured as follows: The literature review explores the evolution of medulloblastoma subgrouping, focusing on molecular subgroup classification and machine learning approaches. The methodology details the data acquisition, quality control measures, metagene extraction, and machine learning model development. The experimental results highlight methylation patterns, subgroup classification performance, and model evaluations. Finally, the discussion interprets the findings, and the conclusion summarizes the study’s contributions and outlines potential future work.

## 2. Literature Review

### 2.1. Molecular Subgroups of Medulloblastoma

Historically, medulloblastoma classification relied solely on histological features, which had clinical limitations. The 2016 revised fourth edition of the WHO Classification of Tumors of the CNS introduced a more precise, layered approach, integrating molecular characteristics alongside histological variants. This update officially incorporated four primary molecular subgroups: WNT (Wingless) activated, SHH (Sonic Hedgehog) activated, and two non-WNT/non-SHH groups, designated as Groups 3 and 4. Within the SHH subgroup, tumors were further divided based on TP53 status, distinguishing between TP53-mutant and TP53-wildtype [8].

The latest fifth edition of the WHO Classification of Tumors of the CNS has refined medulloblastoma classification, identifying new subgroups within the main molecular groups—four within SHH and eight within non-WNT/non-SHH [9]. While this granularity refines molecular subgroups and enhances understanding of the disease’s complexity, its application in targeted therapies and long-term prognostic studies remains the subject of ongoing research [11].

These molecular subgroups of medulloblastoma also exhibit distinct genetic and clinical features, such as age of onset, prognosis, and treatment response. For example, the WNT subgroup has an excellent prognosis, with survival rates over 95% due to its favorable response to standard therapies [12]. In contrast, the SHH subgroup has a variable prognosis, significantly influenced by the patient’s age and TP53 mutation status. TP53 mutations in children aged 3−17 are linked to higher risk and poorer outcomes, while younger patients (<3 years) or those with wild-type TP53 have better prognoses [13]. Groups 3 and 4 generally have worse outcomes, with higher metastasis rates, poor responses to conventional treatments, and increased recurrence risks [14,15]. Group 3, particularly with MYC amplification or metastatic disease, has the worst prognosis [2], while Group 4 is considered low risk if non-metastatic with chromosome 11 loss but high risk if metastasis is present at diagnosis [7]. This variability highlights the crucial role of molecular subgrouping in treatment decisions, emphasizing the need to consider additional factors to optimize outcomes and minimize treatment-related toxicity.

### 2.2. Molecular Subgroup Classification

Various molecular classification techniques have emerged for the clinical classification of medulloblastoma; popular methods include immunohistochemistry (IHC), next-generation sequencing (NGS), and DNA methylation profiling. While IHC has its benefits, it becomes less effective when dealing with tumor heterogeneity and increasing granular subgroups [16]. Although the latest WHO guidelines do not recommend a specific method, NGS and DNA methylation profiling have proven particularly useful in addressing these challenges. NGS excels at providing detailed insights into mutations and genetic alterations, while DNA methylation profiling is highly effective in accurately classifying tumors into molecular subgroups [17,18]. Despite challenges such as high costs and limited availability, these techniques are expected to become cost effective in the long term as they enable for the assessment of multiple molecular parameters required for classification in a single assay [19].

DNA methylation profiling is a technique used to analyze genome-wide DNA methylation patterns, particularly in differentially methylated regions (DMRs) [20]. Methylation profiling commonly employs arrays like Illumina 450K and EPIC (850K) arrays, covering over 450,000 and 850,000 CpG probes, respectively. These arrays provide a comprehensive view of the tumor’s epigenetic landscape, including beta values, which quantify DNA methylation levels on a scale from 0 (unmethylated) to 1 (fully methylated). Identifying DMRs across multiple probes is more robust and replicable than assessing individual CpG sites. By integrating these methylation patterns, clinicians can accurately classify tumors into molecular subgroups [20,21,22]. However, the complexity and volume of data generated require sophisticated analytical tools, leading to the increasing use of machine learning models to classify subgroups and manage the data efficiently.

### 2.3. Machine Learning Approaches

Methylation profiling-based machine learning approaches for medulloblastoma subgroup classification vary significantly across the studies, particularly in the selection of CpG probes for model training, with the numbers ranging from as few as 5 to over 10,000. In a recent study, Rahmani et al. [23] developed prediction models using 399 CpG probes as biomarkers, achieving an average accuracy of 96%, with Support Vector Machines (SVMs) and Artificial Neural Networks (ANN) delivering the best performance. In a related approach, Gómez et al. [24] utilized Linear Discriminant Analysis (LDA) to create two classifiers: one based on six CpG sites to distinguish between WNT, SHH, and non-WNT/non-SHH subgroups, and another using five CpG sites to differentiate between Group 3 and Group 4.

In another study by Schwalbe et al. [25], a methylation-based classifier was developed using a 17 CpG loci signature, implemented through an SVM model, which accurately classified 98% of fresh–frozen samples and 92% of DNA samples derived from formalin-fixed, paraffin-embedded (FFPE) tissue into the four medulloblastoma subgroups. Although not specific to medulloblastoma, Capper et al. [26] utilized up to 10,000 probes to train a Random Forest (RF) model for classifying CNS tumors. In a related study, Alharbi et al. [27] successfully reclassified medulloblastoma tumors that could not be subgrouped using NGS.

As an alternative to methylation-based classification, Hourfar et al. [28] applied machine learning to a reduced feature set of RNA-seq gene expression data, achieving high accuracy in medulloblastoma subgroup prediction. In particular, RF, KNN, and SVM consistently outperformed Decision Tree (DT) and Naive Bayes (NB) classifiers, with accuracies exceeding 90% in several scenarios. A recent review by Hajikarimloo et al. [29] highlighted RF, SVM, and LASSO as the most commonly used classifiers for survival prediction in medulloblastoma, with RF showing strong performance in handling complex interactions and SVM offering good generalization to unseen data. In contrast, deep learning models are predominantly used for image-based classification tasks, such as those involving histopathological analysis in medulloblastoma studies [30].

While these approaches have been effective in classifying the four molecular subgroups of medulloblastoma, they often overlook important factors such as age and risk profiles within subgroups. Moreover, the increasing granularity of medulloblastoma subtypes and the expanding size of methylation arrays present challenges in determining a suitable representation. Choosing too few probes may miss significant methylation patterns, while too many can increase computational demands and potentially reduce model performance. This underscores the need for more efficient approaches in developing subgroup classification models.

Serving as the foundation for this study, Schwalbe et al. [10] identified seven robust and reproducible novel molecular medulloblastoma subgroups by leveraging metagenes, which reflect the underlying methylation patterns across the methylation arrays. These metagenes were derived by applying Non-negative Matrix Factorization (NMF) to the top 10,000 (10K) CpG probes with the highest variance. This study also considered critical factors such as the age of patients within the SHH group and the risk classifications within Group 3 and Group 4. Integrating this metagene-based approach into supervised classification models could potentially capture more complex methylation patterns, enabling even more granular and accurate classifications in the future.

## 3. Methodology

This section outlines the methodology employed to classify molecular subgroups of medulloblastoma using 450K methylation data. As illustrated in Figure 1, the analysis begins with the acquisition of two methylation datasets, GSE93646 and GSE54880, both of which provide processed methylation beta values for analysis. Following data acquisition, a series of quality control and preprocessing steps were carried out to ensure data reliability and integrity. These steps included probe filtering, handling of missing data, and probe selection to enhance signal quality and computational efficiency. After preprocessing, metagene extraction and projection were performed. This involved preparing the primary dataset (GSE93646) for model training and projecting the validation dataset (GSE54880) for evaluation. Several machine learning models were evaluated during the model selection phase to determine the most effective approach. The final classification model was then developed using the selected configuration, and its performance was assessed using standard evaluation metrics.

### 3.1. Data Acquisition

The 450K methylation datasets GSE93646 and GSE54880 were retrieved from the Gene Expression Omnibus (GEO) repository, hosted by the National Center for Biotechnology Information (NCBI). GSE93646 consists of 428 samples and was used as the primary dataset for model training. GSE54880, containing 276 samples, served as an independent validation set to evaluate model performance (Table 1).

Both datasets provide processed beta values, which represent the proportion of DNA methylation at specific CpG probes, ranging from 0 (unmethylated) to 1 (fully methylated). They also include detection *p*-values, indicating the confidence level of each methylation measurement, where lower *p*-values suggest higher reliability.

Each dataset comprises 485,512 rows, corresponding to individual CpG probes. For each sample, two columns are provided: one for the beta value and one for the detection *p*-value. As a result, the primary dataset (GSE93646) contains 856 columns, and the validation dataset (GSE54880) contains 552 columns.

The subgroup classification labels corresponding to the samples in GSE93646 and GSE54880 were obtained from the original publication [10], where the subgroups were initially identified through unsupervised learning methods. In this study, these subgroup labels were used as references for training and evaluating the supervised classification model.

Following metagene extraction and projection, the final dataset consisted of six metagenes, represented as rows, with samples as columns. This transformed format was used as input for the downstream subgroup classification model.

### 3.2. Data Quality Control and Preprocessing

The methylation beta values in the acquired datasets were preprocessed and normalized prior to download. To further refine the data, additional quality control (QC) steps were applied, including probe filtering, missing data handling, and probe selection to improve data reliability and computational efficiency.

Probes mapped to the sex chromosomes (X and Y) were excluded to eliminate potential gender-related biases, based on established genomic annotations. Cross-reactive and multi-mapping probes, which can produce non-specific hybridization signals, were removed to minimize noise and ambiguity [31]. Probes affected by known single nucleotide polymorphisms (SNPs) within 50 base pairs of the target CpG site, with a minor allele frequency (MAF) of 5% or greater, were also excluded to reduce the risk of false methylation measurements [32]. Furthermore, probes with low confidence—defined as having a detection *p*-value above 0.05 or more than 50% missing values across samples—were filtered out. In total, 117,790 unique probes were removed during this process (Table 2).

Samples with non-classifiable subgroup labels were excluded to maintain a consistent supervised learning framework, resulting in a final primary dataset comprising 409 samples.

Following the probe filtering process, the dataset was transposed to align the structure with the input format required by the K-Nearest Neighbors (KNN) algorithm, which expects samples as rows and features (probes) as columns. This restructuring ensured that missing values could be imputed on a per-sample basis, preserving sample-specific variation across the high-dimensional feature space.

Missing values were then imputed using the KNN algorithm, selected for its non-parametric nature, computational efficiency, and ability to preserve local data structures without relying on strong distributional assumptions. Alternative methods, such as Multivariate Imputation by Chained Equations (MICE) and bootstrap Expectation Maximization (EM), were considered; however, these iterative approaches were not adopted due to the high dimensionality of the methylation data and potential convergence issues that could compromise reliability. KNN imputation thus provided a practical and robust solution for this study.

To enhance computational efficiency during model training, the top 10,000 probes with the highest variance across samples were selected. This dimensionality reduction step was performed to retain the most informative features while mitigating computational burden. Following this selection, the dataset was transposed again to match the required input format for NMF, which expects features (probes) as rows and samples as columns.

The same quality control and preprocessing steps—including probe and sample filtering, missing value imputation, variance-based reduction, and transposition where necessary—were applied consistently to both the primary and validation datasets.

### 3.3. Metagene Extraction and Projection

The process begins by extracting metagenes from the preprocessed primary dataset, *P*, and projecting these metagenes onto the preprocessed validation dataset, *V*, to uncover similar methylation patterns for subgroup classification as illustrated in Figure 2 and Figure 3.

To identify methylation patterns and reduce data dimensionality, NMF was applied to *P*. This decomposition produces two matrices: a basis matrix *W_P_*, which represents the contribution of each CpG probe to each metagene, and a coefficient matrix *H_P_*, which reflects how each sample is associated with the metagenes. These metagenes capture the underlying biological patterns present within the data [33,34]. The stability and consistency of the factorization were evaluated using the cophenetic correlation coefficient.

The resulting matrix *H_P_* was then transposed and used to train the subgroup classification model, while the basis matrix *W_P_* was employed for metagene projection onto the validation dataset *V*.

When NMF is applied separately to different datasets, the resulting metagenes often capture dataset-specific structures, which can complicate cross-dataset comparisons. To address this issue, a metagene projection strategy was employed to align the validation dataset *V* with the metagenes extracted from the primary dataset *P*.

This projection process involved estimating a new coefficient matrix *H_V_*, such that the validation data could be approximately represented using the fixed *W_P_* matrix. Although the Moore–Penrose pseudoinverse can be used to estimate *H_V_* [33], this method does not enforce non-negativity, which is essential for preserving the interpretability of methylation data.

To ensure non-negativity, Non-Negative Least Squares (NNLS) was used. The nnls function from the nnls package in R was applied to each individual sample (column) in *V*, using the fixed *W_P_* matrix to estimate the corresponding coefficient vector. By iteratively solving for each sample, the complete coefficient matrix *H_V_* was constructed.

The resulting *H_V_* matrix was then transposed and used to validate the subgroup classification model. This approach enhances consistency, interpretability, and robustness across multiple methylation datasets for the classification of medulloblastoma subgroups.

### 3.4. Classification Model Development

To facilitate the classification model development, the datasets were structured and mapped based on their molecular subgroup labels, which were subsequently encoded. The primary dataset was divided into an 80/20 train test split (train = 329, test = 80), and a validation set (val = 276) was reserved for the final evaluation of model performance. All metagenes were normalized to a range of 0 to 1 to maintain the integrity of methylation intensity, ensuring consistency across the dataset.

For the classification models, four machine learning algorithms were chosen based on their effectiveness in handling complex, high-dimensional datasets: SVM, RF, eXtreme Gradient Boosting (XGB), and KNN. SVM was selected for its ability to separate complex and subtle patterns in high-dimensional spaces, crucial for detecting methylation differences across molecular subgroups. RF was chosen due to its ensemble learning approach and robustness to noise, making it suitable for dealing with noisy methylation features. XGB was included because of its superior performance on structured data problems and its capability to model non-linear interactions between the probes in the metagenes. KNN served as a simpler baseline model, offering a straightforward way to classify samples based on their proximity in the feature space and providing a comparison to more complex models.

Hyperparameter tuning for all models was performed using a randomized grid search combined with a 10-fold cross-validation within the caret package in R to optimize predictive performance (see Table 3 for the hyperparameter tuning grid configurations). The models were trained using the best hyperparameters from Table 4, and their performance was evaluated based on these parameters.

Key evaluation metrics, including F1 score, recall, precision, standard accuracy, and balanced accuracy, were computed as the mean across all classes, with balanced accuracy emphasized as the primary metric to address class imbalance among subgroups (see Table 5 for the class imbalance ratio in the training set). The formulas for calculating these metrics are provided in Table 6. Balanced accuracy was chosen over standard accuracy for this multiclass classification problem with imbalanced classes, as it adjusts for the class distribution and gives a more reliable measure of model performance when there are significant class imbalances.

After comparative evaluation, the SVM model was selected for final validation based on its superior performance. Its reliability was further assessed using confusion matrices, one-vs-all Receiver Operating Characteristic (ROC) curves, and the Area Under the Curve (AUC) metrics.

## 4. Experimentation and Results

### 4.1. Methylation Patterns

The analysis of 450K methylation data, encompassing 485,512 CpG probes, revealed a consistent bimodal distribution of methylation beta values across both the primary and validation datasets, as illustrated in Figure 4 and Figure 5. This distribution, characterized by distinct peaks near 0 and 1, indicated that most CpG probes were either largely unmethylated (hypomethylation) or highly methylated (hypermethylation), suggesting that these methylation states were tightly regulated by underlying biological processes.

While the overall bimodal pattern was consistent between the primary and validation datasets, slight differences in the peak and narrowness of the distribution were observed in the validation data. These discrepancies in the 450K probes could be attributed to variations in the data processing methods used during the preparation of the primary and validation datasets by their respective research teams. Notably, this bimodal distribution persisted and even became more pronounced after applying QC measures and selecting the top 10K CpG probes with the highest variance. However, some deviations from the peaks were observed within the validation data, possibly due to the KNN imputation of missing values.

Despite these minor variations, the clear bimodal distribution likely reflects underlying biological differences between medulloblastoma subgroups, with some characterized by hypermethylation and others by hypomethylation. This persistent pattern underscores its significance in subgroup classification, highlighting its potential as a biomarker for distinguishing between these subgroups.

### 4.2. Metagenes and Medulloblastoma Subgroups

During the metagene extraction process, various numbers of NMF components were tested, and the cophenetic correlation coefficient was calculated to assess the stability of the resulting metagenes. Six metagene components (V1 to V6) were identified as optimal, achieving a cophenetic score of 0.997, consistent with the analysis of the original study carried out by Schwalbe et al. [10].

To further explore the role of these metagenes in subgroup classification, the *H_P_* and *H_V_* matrices were sorted and visualized using heatmaps and t-distributed Stochastic Neighbor Embedding (t-SNE) 2D scatter plots, as shown in Figure 6 and Figure 7, respectively.

The positioning differences in scatter plots between the primary and validation datasets, particularly for the Group 3 and Group 4 subgroups, stem from variations in the t-SNE perplexity parameter used during the visualization process, which influences how the data are represented in the 2D scatter plot. These shifts in the scatter plot do not affect the underlying classification process, as the model relies only on the metagenes.

Despite these differences, the results demonstrated consistent patterns across the metagenes within the seven subgroups in both datasets, validating their significance. Notably, the WNT and SHH subgroups were distinctly separated, while Group 3 and Group 4 showed some overlap, particularly in high-risk and low-risk classifications within and between these subgroups. The overlaps observed at the borders of Group 3 and Group 4 may have had implications for the accuracy of the subgroup classification model, potentially affecting its ability to differentiate these subgroups effectively.

### 4.3. Classification Model Evaluation

During experimentation with the test dataset (Table 7), SVM, RF, and KNN demonstrated strong performance, each achieving a mean F1 score, recall, precision, and standard accuracy of 0.98, along with a mean balanced accuracy of 0.99 across all subgroups. The XGB model, in comparison, exhibited slightly lower performance than the other classification models.

When validated on a larger, independent dataset (Table 8), the SVM model maintained its lead with a mean balanced accuracy of 0.98, closely followed by RF and KNN. The XGB model showed a slight decline in performance, with a mean balanced accuracy of 0.96. Although the mean balanced accuracy of SVM remained high, the standard accuracy dropped slightly from 0.98 on the test data to 0.95 on the validation dation data. This drop, however, remained acceptable and highlights the advantage of emphasizing balanced accuracy for evaluating models on imbalanced multiclass data.

Overall, SVM and RF exhibited the most consistent performance across both test and validation datasets, positioning them as the most reliable models in this analysis. While RF exhibited a slightly faster prediction time (0.0082 seconds) compared to SVM (0.0126 seconds) on the validation set, the predication time was minimal and practically insignificant. Given that SVM consistently showed marginally better-balanced accuracy and overall reliability across datasets, it was selected as the primary method for classifying medulloblastoma subgroups. Further evaluations and analyses were conducted using the SVM model.

Subsequently, confusion matrices were generated (Figure 8) and revealed that the model achieved near-perfect classification of medulloblastoma subgroups, with only a few misclassifications.

In Table 9, alongside the True Label and Predicted Label, two additional metrics are reported: the Predicted True Label Probability and the Predicted Label Probability. These probabilities were calculated by enabling probability estimation during SVM model training using the e1071 package in R. The True Label refers to the actual class assigned to the sample in the dataset, while the Predicted Label is the class predicted by the model. The Predicted True Label Probability represents the probability that the model assigns to the correct class (True Label), while the Predicted Label Probability indicates the probability assigned to the predicted class (the one with the highest probability, Predicted Label). Although these probabilities were computed across all seven subgroups, only the probabilities for the True Label and Predicted label are shown in Table 8 for clarity. For example, in one instance of the test set, the True Label was Group4-LowRisk but was misclassified as Group4-HighRisk, with a probability of 0.30 assigned to the True Label and 0.33 assigned to the Predicted Label, reflecting the model’s uncertainty between these closely related categories.

On the training data, there were 2 misclassifications out of 80 instances, where Group4-LowRisk cases were misclassified as Group4-HighRisk. On the validation data, 12 instances out of 276 were misclassified, primarily within the same subgroup categories. Specifically, two instances of SHH-Infant were incorrectly predicted as SHH-Child, two misclassifications occurred between Group4-HighRisk and Group4-LowRisk, and the highest misclassification rate was observed with 4 Group3-HighRisk instances being misclassified as Group3-LowRisk. These misclassifications are reasonable, given that they occur within the same subgroup categories.

However, a few rare misclassifications across different subgroups were observed: one instance of Group4-HighRisk was misclassified as Group3-HighRisk, and another instance of Group3-HighRisk was misclassified as Group4-LowRisk. These errors could be attributed to overlapping features between the Group 3 and Group 4 categories. The most notable misclassification was one instance of WNT being incorrectly classified as Group3-HighRisk. This misclassification was concerning, as WNT is the most distinct subgroup, despite the prediction having a low confidence probability.

Furthermore, one-vs-all ROC curves were generated (Figure 9) from the validation data, and the macro-averaged AUC across all classes was calculated, reaching an exceptionally high value of 0.9984. This highlights the model’s excellent ability to distinguish between the different medulloblastoma subgroups. This impressive AUC value suggests that the model was highly effective in correctly classifying instances across all subgroups, with very few false positives or negatives. Overall, these results demonstrate the model’s robust performance and reliability in medulloblastoma subgroup classification.

## 5. Discussion

This study demonstrated the effectiveness of using six metagenes derived from 450K methylation data combined with machine learning models for classifying medulloblastoma subgroups. Among the models tested, SVM achieved the highest performance, with a mean balanced accuracy of 98% and a macro-averaged AUC of 0.9984 on an independent validation dataset. This performance is comparable to previous studies, such as those utilizing 399 CpG probes with 99% accuracy on SVM [23] or 17 CpG loci achieving 98% accuracy on SVM [25]. Unlike these studies, which primarily focused on four main subgroups, the current model classifies seven subgroups by incorporating additional factors such as age and risk categories, resulting in more detailed subgroup classification. Moreover, the use of metagenes facilitated the selection of the top 10K probes with the highest variances, thereby broadening the representation of methylation data. This approach not only enhances the coverage of relevant methylation sites but also benefits from dimensional reduction, leading to reduced computational time and resource requirements.

This impressive performance can be attributed to the valuable groundwork laid by the original study by Schwalbe et al. [10], which performed unsupervised subgroup classification incorporating age and risk factors. Their approach led to the identification of seven distinct medulloblastoma subgroups, expanding upon the traditional four-subgroup classification system. The study further established the clinical relevance of these subgroups through Kaplan–Meier survival analysis, which demonstrated substantial differences in 5-year overall survival rates and highlighted the prognostic value of molecular stratification. Specifically, the reported survival rates were WNT (93%), SHH-Infant (62%), SHH-Child (58%), Group3–LowRisk (69%), Group3–HighRisk (37%), Group4–LowRisk (80%), and Group4–High Risk (69%). Although survival analysis could not be conducted in our study due to the lack of survival data in the dataset used, the prognostic relevance of the molecular subgroups classified by our model is supported by these comprehensive findings.

This robust classification framework, with its biologically and clinically defined subgroups, provided a strong foundation for the development of our supervised machine learning model. By leveraging these well-characterized subgroup assignments, our model benefits from both biological significance and clinical relevance, which contributed substantially to its strong performance.

However, it is important to note that the original study, conducted in 2017, was based on the WHO CNS tumors classification fourth edition of medulloblastoma classification, which means the current model does not incorporate the latest WHO CNS tumors classification fifth edition. As the fifth edition offers updated insights into subgroup classification, we plan to integrate these changes into our model in future research, building on the current approach to achieve even greater accuracy and alignment with contemporary clinical standards.

Despite the model’s strong overall performance, some misclassifications were observed, particularly between Group 3 and Group 4 subgroups, as well as occasional errors in other subgroups. These misclassifications could be attributed to discrepancies in probe variance between the training and validation datasets, as well as biases introduced by KNN imputation. Additionally, the higher proportion of missing or low-quality data in the validation set may have contributed to these errors.

To improve classification accuracy and mitigate these issues, future research should focus on refining preprocessing steps and probe selection processes to minimize variance discrepancies and enhance the differentiation between closely related subgroups. This is especially critical as the classification framework evolves to recognize the eight subgroups of Group 3/4 medulloblastoma, as outlined by the WHO and in prominent publications [35]. The shift from high/low-risk variants to principal molecular groups requires a deeper understanding of their implications for survival outcomes. Furthermore, exploring alternative imputation methods and incorporating updated methylation arrays, such as the Illumina EPIC and EPICv2 arrays, could further enhance classification accuracy, as these arrays offer broader coverage and better performance than the discontinued 450K arrays. Validation using publicly available datasets, such as those from [36], could provide a robust framework for comparing the low-risk/high-risk classifications of Group 3/4 with contemporary subgroup assignments.

It is also crucial to emphasize that while these classification results provide significant insights, they should not serve as the sole basis for clinical decision making. Instead, the framework is intended as a supplementary tool, designed to offer valuable recommendations to guide and enhance decision-making processes rather than replace established diagnostic methods.

The web application integrating the classification model is accessible at https://eeecs.shinyapps.io/450k_mb_classification/ (accessed on 6 May 2025).

## 6. Conclusions

This study demonstrated the effectiveness of a supervised machine learning-based approach, utilizing 450K methylation data and leveraging six metagenes, to classify medulloblastoma into seven molecular subgroups with high accuracy. By incorporating age and risk factors, the model offers a more detailed and clinically relevant classification compared to previous approaches, which focused only on four primary subgroups. A series of machine learning models were tested, with the SVM model achieving exceptional performance, attaining 98% mean balanced accuracy and a macro-averaged AUC of 0.99 on an independent validation dataset. The use of metagenes enabled the model to capture a broader representation of methylation data while reducing computational demands, making it both efficient and scalable. With strong potential for clinical application, this model marks a significant advancement in medulloblastoma research and could pave the way for more personalized, subgroup-directed therapies to improve patient outcomes.

## Figures and Tables

**Figure 1 healthcare-13-01114-f001:**
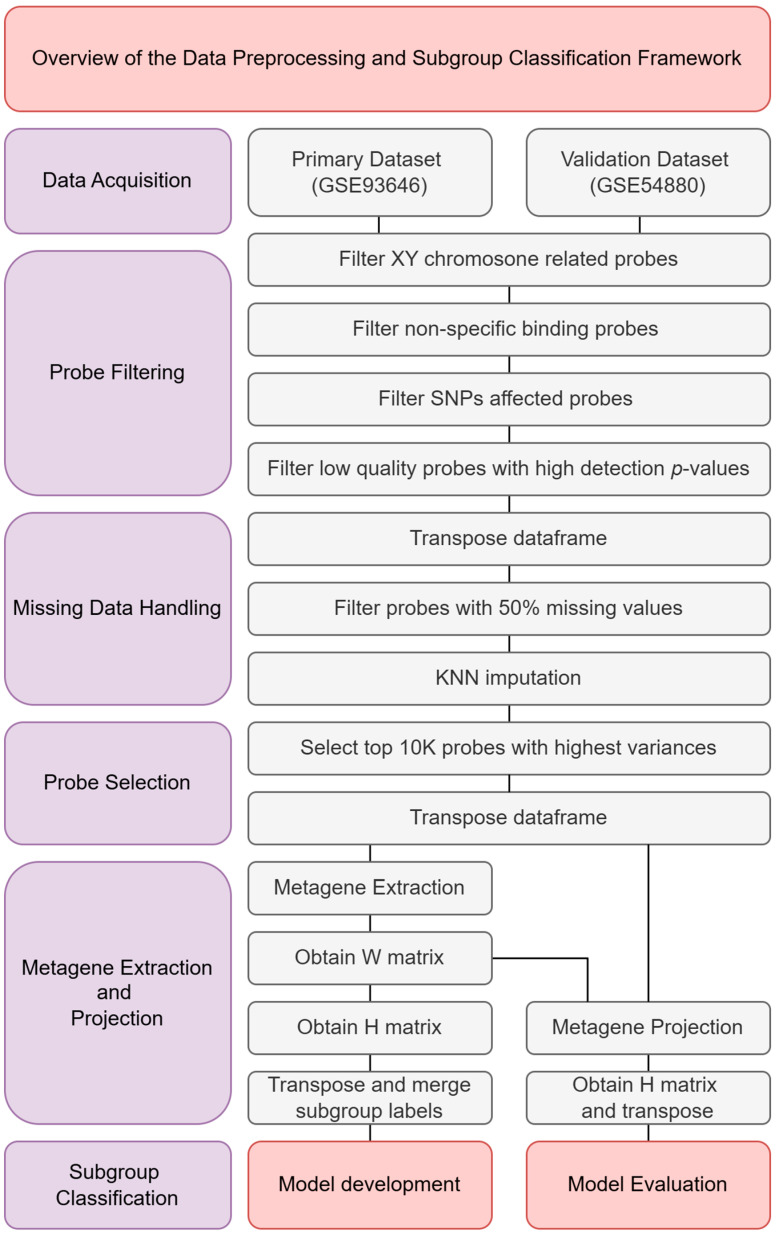
Overview of the data preprocessing and subgroup classification framework.

**Figure 2 healthcare-13-01114-f002:**
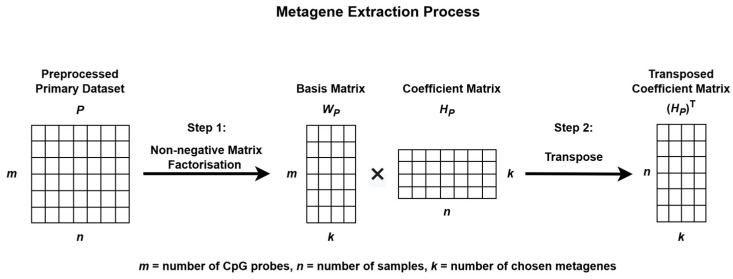
Metagene extraction process using Non-negative Matrix Factorization.

**Figure 3 healthcare-13-01114-f003:**
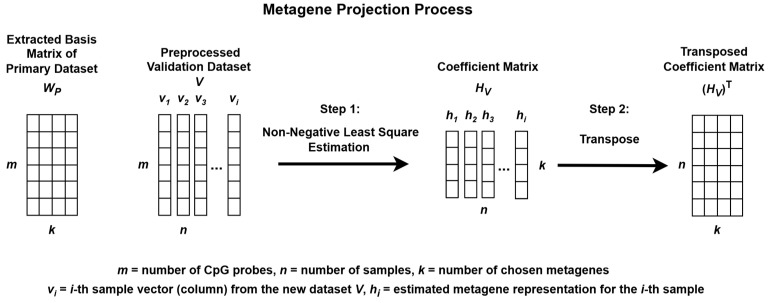
Metagene projection process using Non-negative Least Squares Estimation.

**Figure 4 healthcare-13-01114-f004:**
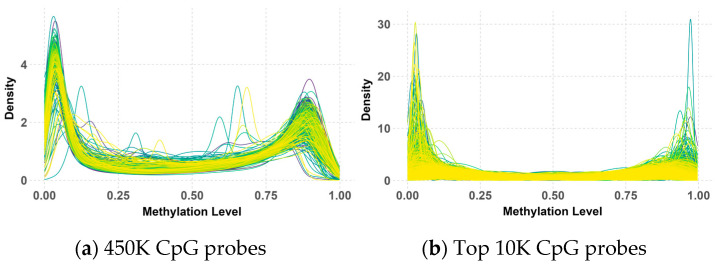
Beta value distribution of 450K (**a**) and top 10K (**b**) CpG probes in the primary dataset.

**Figure 5 healthcare-13-01114-f005:**
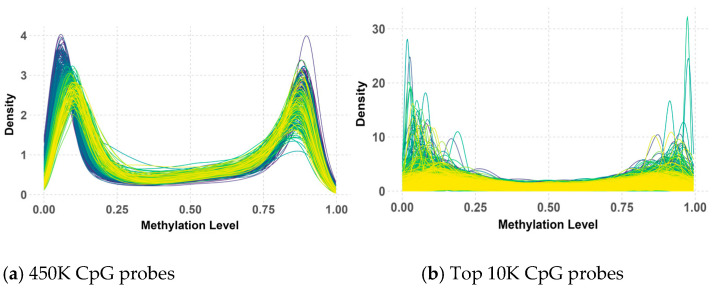
Beta value distribution of 450K (**a**) and top 10K (**b**) CpG probes in the validation dataset.

**Figure 6 healthcare-13-01114-f006:**
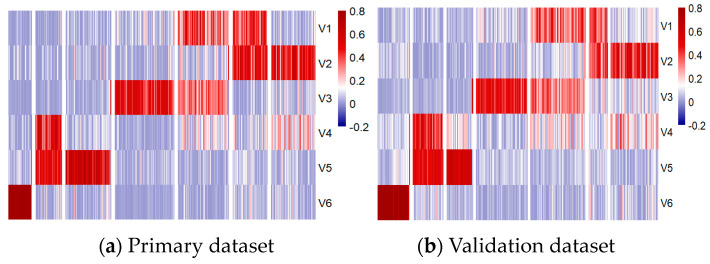
Methylation patterns across six metagenes in primary (**a**) and validation (**b**) datasets.

**Figure 7 healthcare-13-01114-f007:**
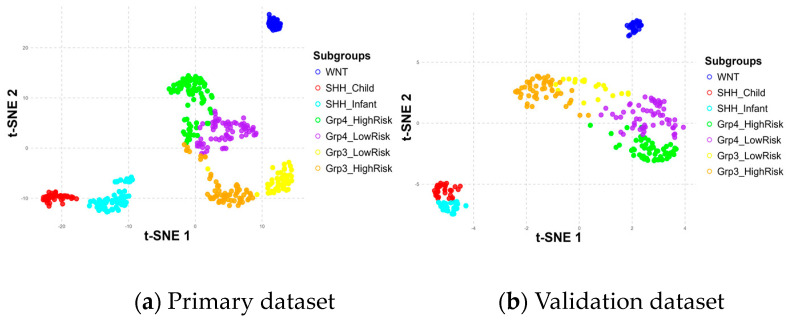
Subgroup distribution across six metagenes in primary (**a**) and validation (**b**) datasets.

**Figure 8 healthcare-13-01114-f008:**
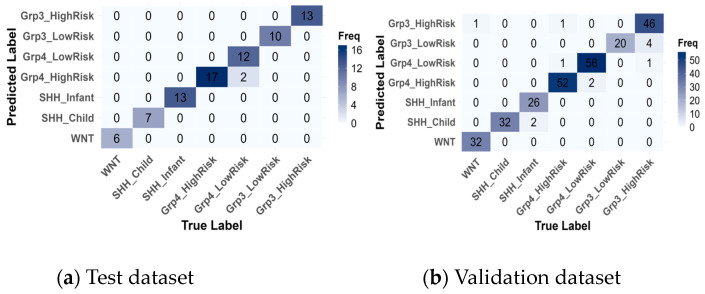
Confusion matrices of the best model, SVM, for the test (**a**) and validation (**b**) datasets.

**Figure 9 healthcare-13-01114-f009:**
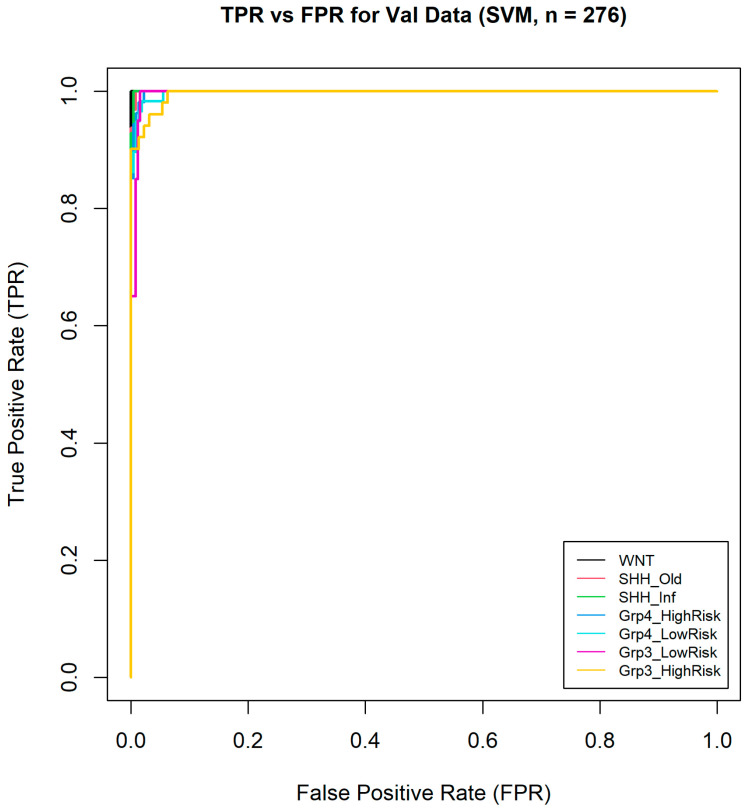
One-vs-all ROC curves of seven medulloblastoma subgroups.

**Table 1 healthcare-13-01114-t001:** Demographics and clinicopathological characteristics of the two utilized datasets [10].

	Primary (n = 428)	Validation (n = 276)
7 subgroup assignment		
WNT	33 (8%)	33 (12%)
SHH-Child	38 (9%)	32 (12%)
SHH-Infant (<4.3 years)	65 (16%)	28 (10%)
Group3-HighRisk	65 (16%)	51 (18%)
Group3-LowRisk	50 (12%)	20 (7%)
Group4-HighRisk	85 (21%)	54 (20%)
Group4-LowRisk	73 (18%)	58 (21%)
Non-classifiable	19	0
Sex		
Male	278 (65 %)	174 (63%)
Female	150 (35%)	102 (37%)
Male–Female ratio	1.9:1	1.7:1
Age at diagnosis (years)		
Median (range)	6.34 (0.24−15.97)	7.50 (0.0−18.0)
<3	101 (24%)	30 (11%)
≥3	327 (76%)	244 (89%)

**Table 2 healthcare-13-01114-t002:** Filtered probes list.

Type	Count
XY mapped	11,648
Non-specific binding	37,639
SNPs affected	59,284
Low confidence	9219
Total unique	117,790

**Table 3 healthcare-13-01114-t003:** Hyperparameters tuning grid configurations for classification models.

Model	Hyperparameter	Range/Values
SVM (Linear)	C	0.01 to 50
SVM (Polynomial)	C	0.01 to 50
	degree	2 to 4
	scale	0.001 to 0.1
SVM (Radial)	C	0.01 to 50
	sigma	0.001 to 1
RF	mtry	2 to 6
XGB	nrounds	50 to 200 (increment by 50)
	eta	0.01 to 0.3
	max_depth	3 to 10
	gamma	0 to 5
	colsample_bytree	0.5 to 1
	min_child_weight	1 to 10
	subsample	0.5 to 1
KNN	k	3 to 15 (odd numbers only)

**Table 4 healthcare-13-01114-t004:** Best tuned hyperparameters for classification models.

Model	Hyperparameter	Range/Values
SVM (Radial)	C	0.61
	sigma	22.54
RF	mtry	2
XGB	nrounds	100
	eta	5
	max_depth	0.14
	gamma	0.02
	colsample_bytree	0.72
	min_child_weight	1.35
	subsample	0.92
KNN	k	13

**Table 5 healthcare-13-01114-t005:** Class imbalance overview in training set among molecular subgroups.

Subgroup Label	Original Sample Count (Primary Dataset)	Training Sample Count	Percentage in Training Set
WNT	33	27	8.2%
SHH-Child	38	31	9.42%
SHH-Infant	65	52	15.81%
Group3-HighRisk	65	52	15.81%
Group3-LowRisk	50	40	12.16%
Group4-HighRisk	85	68	20.67%
Group4-LowRisk	73	59	17.93%
Non-classifiable	19	excluded	excluded
Total	428	329	100%

**Table 6 healthcare-13-01114-t006:** Formulas for calculating performance metrics.

Metric	Formula
Precision	$\frac{True\;Positives}{True\;Positives+False\;Positives}$
Recall	$\frac{True\;Positives}{True\;Positives+False\;Negatives}$
F1 score	$2\times \frac{Precision\times Recall}{Precision+Recall}$
Accuracy	$\frac{True\;Positives+True\;Negatives}{True\;Positives+True\;Negatives+False\;Positives+False\;Negatives}$
Balanced Accuracy	$\frac{1}{N}\sum_{i=1}^{N}\frac{{True\;Positives}_{i}}{{True\;Positives}_{i}+{False\;Negatives}_{i}}$

**Table 7 healthcare-13-01114-t007:** Model performance on test dataset (n = 80).

Model	F1	Recall	Precision	Standard Accuracy	Balanced Accuracy	Prediction Time (s)
SVM	0.98	0.98	0.98	0.98	0.99	0.0077
RF	0.98	0.98	0.98	0.98	0.99	0.0035
XGB	0.96	0.96	0.96	0.95	0.98	0.0070
KNN	0.98	0.98	0.98	0.98	0.99	0.0022

**Table 8 healthcare-13-01114-t008:** Model performance on validation dataset (n = 276).

Model	F1	Recall	Precision	Standard Accuracy	Balanced Accuracy	Prediction Time (s)
SVM	0.95	0.96	0.95	0.95	0.98	0.0126
RF	0.95	0.95	0.94	0.95	0.97	0.0082
XGB	0.92	0.93	0.92	0.93	0.96	0.0047
KNN	0.94	0.94	0.96	0.95	0.96	0.0057

**Table 9 healthcare-13-01114-t009:** Overview of misclassification on test and validation datasets.

Data	True Label [10]	Predicted Label	Predicted True Label Probability	Predicted Label Probability
Test				
1	Grp4-LR	Grp4-HR	0.30	0.33
2	Grp4-LR	Grp4-HR	0.21	0.67
Val				
1	WNT	Grp3-HR	0.09	0.44
2	SHH-Infant	SHH-Child	0.12	0.79
3	SHH-Infant	SHH-Child	0.25	0.66
4	Grp4-HR	Grp4-LR	0.15	0.75
5	Grp4-HR	Grp3-HR	0.22	0.47
6	Grp4-LR	Grp4-HR	0.21	0.73
7	Grp4-LR	Grp4-HR	0.13	0.83
8	Grp3-HR	Grp3-LR	0.09	0.86
9	Grp3-HR	Grp3-LR	0.29	0.53
10	Grp3-HR	Grp4-LR	0.15	0.80
11	Grp3-HR	Grp3-LR	0.10	0.86
12	Grp3-HR	Grp3-LR	0.21	0.74

## Data Availability

The data presented in this study are openly available in the GEO repository: https://www.ncbi.nlm.nih.gov/geo/query/acc.cgi?acc=GSE93646([GEO] accessed on 6 May 2025.

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
