# Peer review of "Integrative Machine Learning Framework for Enhanced Subgroup Classification in Medulloblastoma"

_healthcare, 2025, doi:10.3390/healthcare13101114_

Round 1
Reviewer 1 Report
Comments and Suggestions for Authors
In this research work to classify medulloblastoma into seven molecular subgroups of “WNT, SHH-Infant, SHH-Child, Group3-LowRisk, Group3-HighRisk, Group4-LowRisk, and Group4-HighRisk” of Illumina 450K dataset, a supervised machine learning algorithms SVM, RF, XGB and KNN has been trained. SVM algorithm showed an highest mean balanced accuracy of 98% and AUC score of 99%.
Address the following Questions.
- Under the section 1, provide the Objectives of the study.
- Under the section 3.1, it has been mentioned "Each dataset consists of 485,512 rows corresponding to CpG probes, with columns representing sample IDs and their respective detection p-values". Mention the number of columns per row.
- Under Section 3.2, What is the rationale behind selecting KNN for missing value imputation ? Was any multivariate technique like iterative imputation was experimented?
- Under the section 3.2, Mention what feature selection methods were used during experimentation.
- Under the section 3.4, mention the parameter values used for training the classifications models such as SVM, RF, eXtreme Gradient Boost (XGB), and KNN. For SVM, mention the hyperparameters used for training such as Kernel, Regularization and Gamma value.
- Under the section 3.4, mention class imbalance ratio in the training among the seven molecular subgroups: WNT, SHH-Infant, SHH-Child, Group3-LowRisk, Group3-HighRisk, Group4-LowRisk, and Group4-HighRisk
- In the Abstract and Conclusion section, it has been mentioned that the classification into seven molecular subgroups was carried out by incorporating "age and risk factors". During Machine Learning (ML) model building, were any “feature scaling” techniques have been adapted to maintain common scale of values for “age and risk factors”
- Related to work presented, cite few more references of recent publications. Should take care reference listing format.

Author Response
Comment 1.1: “Under the section 1, provide the Objectives of the study”
Thank you for the helpful suggestion. We have revised the existing text in Section 1 to more clearly state the objective of the study. The improved version now highlights our aim to develop a supervised classification model and software tool for classifying medulloblastoma samples into seven subgroups using metagene methylation patterns. This revision appears in Section 1, Lines 76–83 of the updated manuscript.
Comment 1.2: “Under the section 3.1, it has been mentioned "Each dataset consists of 485,512 rows corresponding to CpG probes, with columns representing sample IDs and their respective detection p-values". Mention the number of columns per row.”
Thank you for your helpful comment. The description under Section 3.1 has been revised to provide a more detailed explanation of the dataset structure. Specifically, the number of columns per dataset has been clearly stated: GSE93646 contains 856 columns (428 samples × 2 columns per sample: beta value and detection p-value), and GSE54880 contains 552 columns (276 samples × 2 columns). This clarification has been incorporated into the revised Data Acquisition section (Section 3.1).
Comment 1.3: “Under Section 3.2, What is the rationale behind selecting KNN for missing value imputation? Was any multivariate technique like iterative imputation was experimented?”
Thank you for your insightful comment. Section 3.2 has been revised to explicitly describe the rationale for selecting K-Nearest Neighbors (KNN) for missing value imputation. As detailed in the updated text, KNN was chosen due to its non-parametric nature, computational efficiency, and its ability to preserve local data structures without assuming an underlying distribution.
Alternative multivariate techniques, including Multivariate Imputation by Chained Equations (MICE) and bootstrap Expectation-Maximization (EM), were also considered. However, these iterative approaches were not adopted due to concerns related to high dimensionality, computational overhead, and potential convergence issues, which could compromise the reliability of imputed values in large-scale methylation datasets. This reasoning is now clearly stated in the revised Section 3.2.
Comment 1.4: “Under the section 3.2, Mention what feature selection methods were used during experimentation.”
Thank you for your comment. To provide greater clarity and avoid potential confusion, the original section title “Quality Control and Feature Selection” has been revised to “Data Quality Control and Preprocessing”. In the context of this study, the term “probe selection” is now used instead of “feature selection” to more accurately describe the process applied to methylation array data.
As detailed in Section 3.2, probe selection was performed by retaining the top 10,000 CpG probes with the highest variance across samples, in order to reduce dimensionality while preserving the most informative signals. These selected probes served as the initial features for downstream analysis. Later in the pipeline, Non-negative Matrix Factorization (NMF) was applied to derive six metagenes, which were then used as features for the final classification model. Both stages of feature representations, both CpG probe-level and metagene-level are now clearly described in the revised methodology.
Comment 1.5: “Under the section 3.4, mention the parameter values used for training the classifications models such as SVM, RF, eXtreme Gradient Boost (XGB), and KNN. For SVM, mention the hyperparameters used for training such as Kernel, Regularization and Gamma value.”
Thank you for your helpful comment. The revised version of Section 3.4 now explicitly refers to the hyperparameter configurations used for model training. Specifically, the parameter tuning ranges for each classification algorithm (SVM, RF, XGB, and KNN) are provided in Table 3, and the optimal hyperparameter values selected during tuning are detailed in Table 4.
For the SVM model, the kernel type, regularization parameter (C), and gamma value used during training are included in Table 4. As the models were implemented using R's caret package, the SVM implementation reflects the underlying configuration of the e1071 package, where gamma and cost are tuned differently compared to other environments (e.g., Python’s scikit-learn). This distinction has been considered and documented to ensure transparency and reproducibility in the classification model development process.
Comment 1.6: “Under the section 3.4, mention class imbalance ratio in the training among the seven molecular subgroups: WNT, SHH-Infant, SHH-Child, Group3-LowRisk, Group3-HighRisk, Group4-LowRisk, and Group4-HighRisk.”
Thank you for your valuable suggestion. To address this, the class distribution within the training set has been calculated and is now presented as percentage ratios in Table 5. The table includes the relative frequency of each molecular subgroup: WNT, SHH-Infant, SHH-Child, Group3-LowRisk, Group3-HighRisk, Group4-LowRisk, and Group4-HighRisk, providing a clear overview of the class imbalance in the training dataset. This information supports the rationale for using balanced accuracy as the primary evaluation metric in this study.
Comment 1.7: “In the Abstract and Conclusion section, it has been mentioned that the classification into seven molecular subgroups was carried out by incorporating "age and risk factors". During Machine Learning (ML) model building, were any “feature scaling” techniques have been adapted to maintain common scale of values for “age and risk factors”.”
Thank you for your comment and the opportunity to clarify this point. The age and risk factors mentioned in the Abstract and Conclusion were not used as input features in the machine learning model. Rather, these factors were incorporated in the original study, Schwalbe et al. [10] during the definition of the seven molecular subgroups, which served as target labels in this work.
In this study, the only features used for model training were derived from methylation beta values of CpG probes, which were then reduced to six metagenes using Non-negative Matrix Factorization (NMF). These metagenes were scaled to a 0–1 range to reflect methylation intensity and ensure consistency across samples.
Therefore, no feature scaling was applied to age or risk factors, as they were not explicitly included in the feature set. Their influence is reflected indirectly, through the subgroup labels obtained from the original classification.
Comment 1.8: “Related to work presented, cite few more references of recent publications. Should take care reference listing format.”
Thank you for your helpful suggestion. In response, additional recent studies have been incorporated into the literature review under the "Machine Learning–Based Approaches" section. Specifically, we have added:
- Hourfar et al. [28], which explores RNA-seq–based ML classification of medulloblastoma subgroups (suggested by Reviewer 2),
- Hajikarimloo et al. [29], a 2025 systematic review highlighting the most frequently used classifiers for survival prediction in medulloblastoma (also suggested by Reviewer 2), and
- Attallah and Zaghlool [30], which represents a recent deep learning approach used for image-based medulloblastoma classification.
These references expand the context of our study and support the rationale behind our model choices. The citations have been formatted according to the IEEE reference style, as requested.

Reviewer 2 Report
Comments and Suggestions for Authors
-
Lines 36-39 gives a comparison of children having CNS tumor in UK and USA. It is better to represent the comparison in same unit. i.e. either number of cases or percentage in both cases to improve readability.
-
What is "CpG probes"? It should be explained before the first mention of it.
-
On line 145, "For example, one study developed prediction models using " should be more specific to the author name or method name instead of one study. Literature review should be with respect to the authors or models used in those studies.
-
More machine learning-based literature review needs to be added in this manuscript such as:
-
Machine Learning–Driven Identification of Molecular Subgroups in Medulloblastoma via Gene Expression Profiling
-
MS-DLD: Multi-Sensors Based Daily Locomotion Detection via Kinematic-Static Energy and Body-Specific HMMs
-
Machine learning-based models for prediction of survival in medulloblastoma: a systematic review and meta-analysis
-
- A general idea about methodology should be given in the form of text and a figure before Data Acquisition section.
- Figure 1 needs more explanation.
- There is no explanation about models trained in the manuscript. Hyperparameters tuned also should be mentioned in a table format for each model used. A brief explanation of each model along with why they were used for this problem should be included.
- How the different evaluation metrics were calculated in section 4.3? How much time was taken by each utilized model to predict?
- Explain the true label probability and label probability in Table 5. How were they calculated?
- The proposed method achieved exceptionally higher results as mentioned by the authors themselves. What are the reasons behind achieving such high outcomes?
- Manuscript format needs to be checked thoroughly.
- More figures should be added to enhance readability.
Author Response
Comment 2.1: “Lines 36-39 gives a comparison of children having CNS tumor in UK and USA. It is better to represent the comparison in same unit. i.e. either number of cases or percentage in both cases to improve readability.
Thank you for this valuable comment. We have revised the sentence to ensure consistency in the units used for comparison. Both statistics are now presented as incidence rates per 100,000 person-years, improving readability and comparability. The revised text appears in Section 1, Lines 36–39 of the updated manuscript.
Comment 2.2: “What is "CpG probes"? It should be explained before the first mention of it.”
Thank you for your helpful suggestion. We have revised the relevant section to provide a clear explanation of CpG sites and CpG probes before their first mention. Specifically, we now define CpG as a DNA sequence where a cytosine is followed by a guanine, and explain that CpG probes are synthetic DNA sequences designed to detect methylation at these sites. This clarification improves the accessibility of the text for readers unfamiliar with epigenetic terminology. The updated content appears in Section 1, Lines 60–69 of the revised manuscript.
Comment 2.3: “On line 145 “For example, one study developed prediction models using " should be more specific to the author name or method name instead of one study. Literature review should be with respect to the authors or models used in those studies.””
Thank you for this valuable comment. We have replaced vague references with specific author names and models where applicable in the revised manuscript.
Comment 2.4: “More machine learning-based literature review needs to be added in this manuscript such as:
- Machine Learning–Driven Identification of Molecular Subgroups in Medulloblastoma via Gene Expression Profiling
- MS-DLD: Multi-Sensors Based Daily Locomotion Detection via Kinematic-Static Energy and Body-Specific HMMs
- Machine learning-based models for prediction of survival in medulloblastoma: a systematic review and meta-analysis”
Thank you for the valuable suggestions. In response, we have incorporated the first and third studies into the literature review section on Section 2.3 Machine Learning Based Approaches. Specifically, Hourfar et al. [28] and Hajikarimloo et al. [29] have been cited to support recent advances in gene expression–based subgroup classification and survival prediction in medulloblastoma, respectively. These studies also reinforce our model selection strategy, as both identified RF and SVM as high-performing classifiers, with RF offering strong handling of complex interactions and SVM demonstrating robust generalization attributes well-suited to high-dimensional methylation data.
The second suggested study “MS-DLD: Multi-Sensors Based Daily Locomotion Detection via Kinematic-Static Energy and Body-Specific HMMs” focuses on sensor-based human activity recognition and does not align with the biomedical data or objectives of this work. Therefore, instead of this reference, we have included a more relevant study by Attallah and Zaghlool [30], which applies deep learning to image-based medulloblastoma classification. This addition highlights the modality-specific applications of machine learning and helps contrast our methylation-based approach with image-driven models.
Comment 2.5: “A general idea about methodology should be given in the form of text and a figure before Data Acquisition section.”
Thank you for this valuable suggestion. In response, an introductory overview of the methodology has been added prior to the Data Acquisition section. This provides a high-level summary of the overall analytical framework used in the study. Additionally, Figure 1 has been included to visually present the key methodological steps, including data acquisition, preprocessing, metagene extraction and projection, and model development.
Comment 2.6: “Figure 1 needs more explanation.”
Thank you for your helpful feedback. In response, the original Figure 1 has been replaced with two separate diagrams, Figure 2 and Figure 3 to improve clarity and provide a more detailed illustration of the methodological steps. The accompanying explanation in Section 3.3 (Metagene Extraction and Projection) has also been significantly revised to describe each step of the process in detail, including the use of Non-negative Matrix Factorization (NMF), the rationale for metagene projection, and the application of Non-Negative Least Squares (NNLS). These updates aim to enhance the overall interpretability and consistency of the workflow, particularly in the context of cross-dataset subgroup classification.
Comment 2.7: “There is no explanation about models trained in the manuscript. Hyperparameters tuned also should be mentioned in a table format for each model used. A brief explanation of each model along with why they were used for this problem should be included.”
Thank you for your constructive feedback. In response, Section 3.4 (Classification Model Development) has been thoroughly revised to include a detailed explanation of each machine learning model used in this study – SVM, RF, XGB, KNN. A brief rationale is provided for each model, outlining its suitability for handling high-dimensional and complex methylation data.
Additionally, information on hyperparameter tuning has been incorporated. The parameter tuning ranges are presented in Table 3, and the optimal hyperparameter values used for final model training are detailed in Table 4. These updates aim to improve the transparency and reproducibility of the model selection and evaluation process.
Comment 2.8: “How the different evaluation metrics were calculated in section 4.3? How much time was taken by each utilized model to predict?”
Thank you for your comment. In response, Section 3.4 has been revised to clarify the calculation of evaluation metrics used to assess model performance. The formulas for each metric (including accuracy, balanced accuracy, F1 score, precision, and recall) are now provided in Table 6 for transparency.
In addition, the prediction times for each classification model have been included in Tables 7 and 8, which present detailed performance comparisons. The prediction times ranged from 0.0022 to 0.0126 seconds, highlighting the computational efficiency of all models evaluated.
The rationale for choosing these metrics, particularly balanced accuracy has also been discussed to reflect the class imbalance in the molecular subgroup distribution.
Comment 2.9: “Explain the true label probability and label probability in Table 5. How were they calculated?”
Thank you for your comment. The explanation for the Predicted True Label Probability and Predicted Label Probability has been added to Section 4.3. These probabilities were calculated by enabling probability estimation during training of the SVM model using the e1071 package in R.
The Predicted True Label Probability refers to the probability that the model assigns to the actual (true) class of the sample, while the Predicted Label Probability is the probability assigned to the predicted class (i.e., the class with the highest predicted probability). These values offer insight into the model’s confidence and are particularly useful for interpreting misclassifications.
Although probabilities were generated for all seven classes, only the two relevant values (for the true and predicted labels) are reported in Table 9 for clarity. An example has also been provided in Section 4.3 to illustrate how these values reflect the model's uncertainty in cases of misclassification
Comment 2.10: “The proposed method achieved exceptionally higher results as mentioned by the authors themselves. What are the reasons behind achieving such high outcomes?”
Thank you for your thoughtful comment. The reasons for the high classification performance have been discussed in the Discussion section. Specifically, the proposed method builds upon the robust framework established by Schwalbe et al. [10], whose unsupervised classification incorporated age and risk factors to define seven clinically meaningful subgroups. These subgroup labels provided a strong biological and clinical basis for training our supervised model.
Comment 2.11: “Manuscript format needs to be checked thoroughly.”
Thank you for your feedback. The manuscript has been thoroughly reviewed and revised to ensure consistency with the formatting and style guidelines of the journal. Attention has been given to section headings, figure and table formatting, citation style, and overall structure to improve readability and presentation.
Comment 2.12: “More figures should be added to enhance readability.”
Thank you for the suggestion. In response, additional figures have been included to improve the overall readability and visual representation of the methodology and results. The original figures were redrawn for clarity, and new diagrams (e.g., updated framework, metagene extraction and projection) have been added to illustrate key processes. Supporting tables and metric formulas have also been incorporated to aid interpretation of model performance and evaluation. These revisions aim to enhance both clarity and comprehension for the reader.

Reviewer 3 Report
Comments and Suggestions for Authors
The paper is titled as "Integrative Machine Learning Framework for Enhanced Subgroup Classification in Medulloblastoma," and it is aimed to develop a supervised machine learning-based approach using DNA methylation data (Illumina 450K methylation arrays) to accurately classify medulloblastoma tumors into seven molecular subgroups, enhancing precision in subgroup-directed therapies and clinical outcomes.
The topic is interesting and fits in the scope of Healthcare journal.
However, I have some corrections and suggestions to improve the paper before acceptance.
1)
Figure 1 (Metagene extraction and projection pipeline) is unclear and should be redrawn.
2)
"I would like to understand why the distributions of the Primary Dataset and the Validation Dataset differ.
For instance, the green color (Grp4) appears in different locations in these two datasets. Why does this discrepancy occur?
Additionally, how can accurate classification be achieved under these circumstances?"
3)
In the paper, many statistical analyses rely on Balanced Accuracy; however, its formulation has not been provided
4)
"Table 3 (Model performance on the test dataset, n=80)" presents valuable comparative results.
However, I would like to see the standard accuracy values included as well.
It is important to observe the difference between standard accuracy and balanced accuracy.
5)
Based on 'Figure 6: Confusion matrices of the best model (SVM) for the test (a) and validation (b) datasets,' as well as 'Table 3: Model performance on the test dataset (n=80)'
and 'Table 4: Model performance on the validation dataset (n=276),'
there appear to be some inconsistencies.
Please re-check these results for accuracy.
6) I would like to see more detailed information about the dataset.
Specifically, how many features were used, and what exactly are these features?
In data science, these details are essential for understanding and interpreting the results.
Author Response
Comment 3.1: “Figure 1 (Metagene extraction and projection pipeline) is unclear and should be redrawn.”
Thank you for your helpful feedback. In response, the original Figure 1 has been redrawn and reorganized to improve clarity and structure. The updated Figure 1 now presents the full methodological framework of the study, providing a high-level overview of the entire pipeline from data acquisition to model evaluation.
To further improve readability and detail, the original content related to metagene extraction and projection has been expanded and divided into two separate illustrations, now presented as Figure 2 and Figure 3. These figures provide a clear, step-by-step visual representation of the metagene extraction using NMF and the projection strategy using NNLS. Additionally, the accompanying explanation in Section 3.3 (Metagene Extraction and Projection) has been thoroughly revised to describe these steps with greater detail and clarity. These revisions aim to enhance the interpretability of the workflow, particularly in the context of cross-dataset analysis and subgroup classification.
Comment 3.2: "I would like to understand why the distributions of the Primary Dataset and the Validation Dataset differ. For instance, the green color (Grp4) appears in different locations in these two datasets. Why does this discrepancy occur? Additionally, how can accurate classification be achieved under these circumstances?"
Thank you for your thoughtful comment. The variation in the placement of subgroups particularly Group 3 and Group 4 between the Primary and Validation datasets in the t-SNE plots arises from differences in the perplexity parameter used during the visualization process. t-SNE is a non-linear dimensionality reduction technique, and its output can vary depending on parameter settings, especially perplexity, which affects how local and global relationships are represented in 2D space.
These differences in the visual arrangement of clusters do not influence the classification performance, as the machine learning models are trained and evaluated using the metagene features, not the t-SNE projections. Classification is based on the high-dimensional metagene space, ensuring consistent subgroup prediction regardless of variations in 2D visualization.
This explanation has been included in Section 4.2 (Metagenes and Medulloblastoma Subgroups) to clarify the distinction between visualization output and the actual classification process.
Comment 3.3: “In the paper, many statistical analyses rely on Balanced Accuracy; however, its formulation has not been provided”
Thank you for your observation. The formulation of balanced accuracy has now been explicitly included in Table 6, along with the definitions of other evaluation metrics. Additionally, the rationale for selecting balanced accuracy as the primary performance metric particularly due to class imbalance in the dataset has been discussed in Section 3.4. These updates aim to improve transparency and ensure clarity regarding the statistical analyses used in the study.
Comment 3.4: “Table 3 (Model performance on the test dataset, n=80)" presents valuable comparative results. However, I would like to see the standard accuracy values included as well. It is important to observe the difference between standard accuracy and balanced accuracy.”
Thank you for your insightful comment. In the revised manuscript, both standard accuracy and balanced accuracy have been reported in the performance comparison tables (see Tables 7 and 8). The differences between these metrics especially in the context of class imbalance have been further addressed in Section 3.4 and Section 4.3. This comparison helps clarify the importance of balanced accuracy for fair model evaluation across underrepresented subgroups.
Comment 3.5: “Based on 'Figure 6: Confusion matrices of the best model (SVM) for the test (a) and validation (b) datasets,' as well as 'Table 3: Model performance on the test dataset (n=80)' and 'Table 4: Model performance on the validation dataset (n=276),' there appear to be some inconsistencies. Please re-check these results for accuracy. “
Thank you for your careful observation. The results presented in Figure 8 (previously Figure 6) and Tables 7 and 8 (previously Tables 3 and 4) are consistent when interpreted using balanced accuracy, which was selected as the primary evaluation metric due to the class imbalance across molecular subgroups.
The confusion may have arisen from differences observed when using standard accuracy, which tends to favour majority classes and can therefore appear inconsistent in imbalanced datasets. In contrast, balanced accuracy accounts for recall across all classes and provides a more representative measure of model performance.
The relevant formulas for balanced accuracy and other metrics have been clearly provided in Table 6, and the rationale for its use has been discussed in Section 3.4 and Section 4.3. We have re-checked all calculations and confirmed the consistency of results when using balanced accuracy as the evaluation standard.
Comment 3.6: “I would like to see more detailed information about the dataset. Specifically, how many features were used, and what exactly are these features? In data science, these details are essential for understanding and interpreting the results.”
Thank you for your insightful comment. The Data Acquisition section (Section 3.1) has been revised to clearly describe the nature and number of features used in this study. Initially, each dataset contained 485,512 CpG probes, each represented by a methylation beta value ranging from 0 (unmethylated) to 1 (fully methylated). These beta values served as the initial feature set.
To reduce dimensionality and enhance computational efficiency, the top 10,000 probes with the highest variance across samples were selected. These high-variance CpG probes formed the input for Non-negative Matrix Factorization (NMF), which was used to extract six metagenes—representing the final features used for model training and evaluation. This process is now described in detail in Sections 3.1 and 3.3, and summarized visually in Figure 1, 2 and 3.
Reviewer 4 Report
Comments and Suggestions for Authors
Comments and Suggestions for the manuscript of healthcare-3575408:
In this manuscript, the authors address a critical need in pediatric neuro-oncology by developing a machine learning framework to classify medulloblastoma into seven molecular subgroups using DNA methylation data. The study is innovative in integrating age and risk factors into the classification model, which enhances subgroup differentiation beyond the traditional four subgroups. Employing metagenes derived from the top 10,000 probes with the highest variance is a novel strategy that reduces computational demands while improving classification accuracy. This method holds significant potential for enhancing treatment strategies and patient outcomes.
Suggestions:
--The authors should map their seven molecular subgroups (WNT, SHH-Infant, SHH-Child, Group 3-Low Risk, Group 3-High Risk, Group 4-Low Risk, Group 4-High Risk) to the corresponding subgroups defined in the WHO 5th edition, particularly the SHH subtype. This mapping would demonstrate how well the model aligns with established clinical guidelines, thereby enhancing its credibility and potential for clinical application.
--The authors could include and compare survival outcomes, such as Kaplan–Meier survival curves, along with clinical features associated with each subgroup. This addition would help validate the model's predictive power.
--If possible, the authors should compare the molecular metagene markers used in the model with those highlighted in the WHO 5th edition. Discussing these markers in relation to traditional classification could provide enhanced clinical benefits for subgroup classification.
--The authors should address any discrepancies found between the model and the WHO classification (5th edition). Highlighting how the model's approach might offer advantages or additional insights compared to the WHO criteria would be beneficial.
Author Response
Comment 4.1: “The authors should map their seven molecular subgroups (WNT, SHH-Infant, SHH-Child, Group 3-Low Risk, Group 3-High Risk, Group 4-Low Risk, Group 4-High Risk) to the corresponding subgroups defined in the WHO 5th edition, particularly the SHH subtype. This mapping would demonstrate how well the model aligns with established clinical guidelines, thereby enhancing its credibility and potential for clinical application.”
Thank you for this thoughtful and clinically relevant suggestion. The current study is based on the molecular subgroup definitions proposed by Schwalbe et al. (2017) [10], which align with the WHO 4th edition classification of medulloblastoma. These include seven subgroups, incorporating age and risk stratification within the SHH, Group 3, and Group 4 categories.
We acknowledge the refinements introduced in the WHO5, which further subclassifies SHH, Group 3, and Group 4 medulloblastomas based on molecular characteristics such as TP53 mutation status and DNA methylation profiles. However, a direct mapping of our seven subgroups particularly for the SHH subtype to the updated WHO5 categories is not currently feasible without reprocessing the dataset using updated clinical annotations and molecular labels.
As noted in the revised Discussion section, integrating WHO5-defined subgroup classifications is an important direction for future work. Doing so will further enhance the clinical relevance of the model and improve its alignment with contemporary diagnostic and treatment standards.
Comment 4.2: “The authors could include and compare survival outcomes, such as Kaplan-Meier survival curves, along with clinical features associated with each subgroup. This addition would help validate the model's predictive power.”
Thank you for this valuable suggestion. We agree that comparing survival outcomes, such as Kaplan-Meier survival curves, would further validate the predictive power and clinical relevance of the proposed classification model. In the present study, the subgroup labels used for training and evaluation were obtained from Schwalbe et al. (2017) [10], who conducted extensive survival analyses including Kaplan-Meier analysis during their original unsupervised classification. These analyses confirmed the prognostic significance of the seven subgroups, particularly with respect to age and risk stratification.
While we did not replicate survival analysis within this study due to the lack of individual-level survival annotations, we recognize its importance. Future extensions of this work will aim to incorporate clinical metadata and survival outcomes, enabling a more comprehensive validation of the model's predictive capabilities within a clinical framework.
Comment 4.3: “If possible, the authors should compare the molecular metagene markers used in the model with those highlighted in the WHO 5th edition. Discussing these markers in relation to traditional classification could provide enhanced clinical benefits for subgroup classification.”
Thank you for your insightful suggestion. The metagenes used in this study were derived using Non-negative Matrix Factorization (NMF) applied to high-variance CpG methylation probes, allowing for an unsupervised extraction of underlying methylation patterns that define subgroup structure. These metagenes represent latent features and do not directly correspond to predefined molecular markers such as MYC amplification, TP53 mutations, or CTNNB1 status, which are central to the WHO 5th edition classification.
At present, a direct comparison between our model’s metagenes and WHO5-defined biomarkers is not feasible due to the data-driven nature of the approach and the lack of explicit biological annotation for each metagene. However, we agree that interpreting and mapping metagenes to known clinical or molecular markers could significantly improve the clinical interpretability and utility of the model.
This will be a key focus of future work, where we aim to incorporate additional genomic annotations and perform a biological characterization of metagenes, potentially aligning them with known prognostic and diagnostic biomarkers outlined in WHO5.
Comment 4.4: “The authors should address any discrepancies found between the model and the WHO classification (5th edition). Highlighting how the model's approach might offer advantages or additional insights compared to the WHO criteria would be beneficial.”
Thank you for this thoughtful comment. The current study is based on the molecular subgroup definitions proposed by Schwalbe et al. (2017) [10], which reflect the WHO4 classification and include additional risk stratification (high/low risk) within Group 3 and Group 4 subtypes. In contrast, the WHO5 redefines Group 3 and Group 4 into eight refined molecular subgroups, while placing greater emphasis on specific molecular alterations such as MYC/MYCN amplification, TP53 status, and chromosomal aberrations.
Due to the differences in classification structure, a direct one-to-one mapping between our model and WHO5-defined subgroups is not currently possible. However, our model offers a risk-stratified view that may still hold clinical value, especially for settings where full genomic annotation is unavailable. Additionally, the use of data-driven metagenes derived from methylation patterns allows the model to potentially capture latent subgroup heterogeneity that may complement or extend beyond predefined biomarker-based groupings.
These differences are now acknowledged and discussed in the Discussion section. Future work will focus on aligning the model with WHO5 subgroup definitions, including the integration of known biomarkers and subgroup-specific genomic features, while preserving the interpretability and predictive strength of the current metagene-based framework.
Round 2
Reviewer 2 Report
Comments and Suggestions for Authors
All the comments have been addressed by the authors.
Author Response
We sincerely thank the reviewer for their positive evaluation and for accepting our revised manuscript. We appreciate the time and effort dedicated to reviewing our work and are grateful for the constructive feedback provided during the review process.
Reviewer 3 Report
Comments and Suggestions for Authors
The authors have carefully considered and addressed the concerns raised in the previous round of reviews. The revised manuscript reflects substantial improvements in clarity, structure, and scientific rigor. All major and minor issues, including those related to methodology, literature support, data interpretation, and presentation quality, appear to have been resolved satisfactorily.
It can be accepted as is.
Author Response

(The authors gave the same response as above.)

Reviewer 4 Report
Comments and Suggestions for Authors
Comments and Suggestions for the manuscript of healthcare-3575408R2:
In this manuscript, the authors address a critical need in pediatric neuro-oncology by developing a machine learning framework to classify medulloblastoma into seven molecular subgroups using DNA methylation data. The study is innovative in integrating age and risk factors into the classification model, which enhances subgroup differentiation beyond the traditional four subgroups. Employing metagenes derived from the top 10,000 probes with the highest variance is a novel strategy that reduces computational demands while improving classification accuracy. This method holds significant potential for enhancing treatment strategies and patient outcomes.
Suggestions:
-- It is reasonable and acceptable that authors can’t perform and compare their classification with the classification by the WHO 5th edition. Importantly, the authors also addressed this issue in this version.
-- -- After the patients were classified by the DNA methylation data, the authors should know the patient survival times based on this classification so that the Kaplan–Meier survival curves for this classification could be obtained. It would greatly enhance your manuscript to add these survival curves based on your classification. Considering that medulloblastoma is a pediatric tumor, it might be difficult to follow patient survival.
Author Response
Reviewer comment: -- It is reasonable and acceptable that authors can’t perform and compare their classification with the classification by the WHO 5th edition. Importantly, the authors also addressed this issue in this version.
-- -- After the patients were classified by the DNA methylation data, the authors should know the patient survival times based on this classification so that the Kaplan–Meier survival curves for this classification could be obtained. It would greatly enhance your manuscript to add these survival curves based on your classification. Considering that medulloblastoma is a paediatric tumour, it might be difficult to follow patient survival.
Response:
we sincerely thank you for your thoughtful comments and for understanding the limitations of our study.
We appreciate your acknowledgment regarding the WHO 5th edition classification. This point was addressed in the first revision, and we are grateful for your understanding.
Regarding the Kaplan–Meier survival analysis, we fully agree that including survival curves would enhance the clinical relevance of our classification model. However, survival data were not available in the processed version of the dataset we used for this study, which made it impossible to perform direct Kaplan–Meier analysis.
To address this, we have referenced the study by Schwalbe et al. [10], which defined the same seven molecular subgroups and performed a comprehensive survival analysis. Their reported 5-year overall survival rates ranging from 93% for WNT to 37% for Group 3–High Risk are now included in the revised Discussion section to provide readers with a clearer understanding of the clinical relevance of these subgroups. All corresponding changes have been highlighted in blue for your convenience.
In future work, we plan to extend this study to incorporate the latest WHO CNS 5th edition classification and to conduct survival analysis using datasets that include matched clinical outcome data.
Thank you again for your valuable feedback.